# Prevalence and Frequency of Non-Fatal Workplace Injuries Among Waste Recyclers at Buy-Back Centres in Johannesburg, South Africa: A Cross-Sectional Study

**DOI:** 10.3390/ijerph22091348

**Published:** 2025-08-28

**Authors:** Hlologelo Ramatsoma, Melitah Motlhale, Thulani Moiane, Kerry Wilson, Nisha Naicker

**Affiliations:** 1Epidemiology and Surveillance Section, National Health Laboratory Service, National Institute for Occupational Health, Johannesburg 2001, South Africa; hlologelor@nioh.ac.za (H.R.); melitahm@nioh.ac.za (M.M.); thulanim@nicd.ac.za (T.M.); kerryw@nioh.ac.za (K.W.); 2School of Public Health, Faculty of Health Sciences, University of the Witwatersrand, Johannesburg 2001, South Africa; 3Department of Environmental Health, University of Johannesburg, Johannesburg 2001, South Africa

**Keywords:** waste recyclers, occupational health, occupational injuries, workplace injuries, frequency of injuries

## Abstract

Physical hazards are the most common source of health effects among waste recyclers, frequently leading to worker injuries. South Africa’s formal buy-back centres (BBCs) have emerged as key nodes in the recycling chain, yet the burden of non-fatal workplace injuries among BBC recyclers is not characterised. We conducted a cross-sectional study at ten BBCs in Johannesburg, enrolling 160 waste recyclers (median age 32 years; 55.6% female). A structured, interviewer-led questionnaire captured workers’ characteristics and self-reported injuries in the past six months. Robust Poisson regression was fitted to determine associations with frequent workplace injury. Overall, 69.4% of participants reported at least one injury. Cuts and lacerations (67.6%) and sprains or muscle strains (39.6%) predominated. Each additional year of age raised the risk of frequent workplace injury by 1% (adjusted relative risk [aRR] 1.01; 95% CI 1.00–1.02), each extra hour worked per day by 22% (aRR 1.22; 95% CI 1.04–1.42), and presence of hearing or vision problems by 45% (aRR 1.45; 95% CI 1.14–1.83). Targeted interventions—such as work hour regulation, sensory-friendly accommodations, and comprehensive, fit-focused PPE programs—are needed to reduce injury risk in this vulnerable workforce.

## 1. Introduction

South Africa, a low to middle-income country, is experiencing a sharp increase in solid waste generation due to increased urbanisation and population growth [1]. Gauteng, the province with the largest population and highest industrial density, contributes the largest share of waste in South Africa, producing over five million tonnes annually [2]. Of the six districts in Gauteng, Johannesburg—the largest city in South Africa—generates a significant portion of domestic waste, approximately 1.4 million tonnes per year, most of which ends up in landfills. The city is facing a growing challenge, as three of its four landfill sites are nearing capacity, with less than 10 years of disposal space remaining [2]. This increase in waste volume has created economic opportunities for both formal recycling firms and informal waste pickers, supported by national policies that prioritise waste prevention, reuse, recycling, and resource recovery [3].

An estimated 15 million people in middle-income countries sustain their livelihoods through waste recycling, separating and selling recyclable materials recovered from municipal solid waste [4]. In South Africa, waste recycling involves a multi-stage process. It starts with separating recyclable materials from general waste, often done by informal “landfill recyclers”, “street recyclers”, or “waste pickers”. These recyclables are then collected, sometimes through intermediaries (“middlemen”), and sold to buy-back centres (BBCs), which purchase large quantities of semi-separated recyclables. These centres then prepare the materials for processing into raw materials used in manufacturing [5,6,7].

The tasks of waste recyclers at BBCs include sorting using hands or machines, cleaning, and feeding recyclables into a baling machine [5,8]. During these practices, waste recyclers are indisputably exposed to ergonomic, biological, physical, and chemical occupational-related hazards [9]. Previous research has reported an association between working at recycling sites and increased poor health problems and injuries [10]. Physical hazards are the most common source of health effects in waste recycling, frequently resulting in worker injuries [9]. A cross-sectional study conducted among solid waste management workers reported a prevalence of occupational-related injuries of 73.2% [11]. Common injuries include cuts and punctures to workers’ hands, arms, and legs from handling sharp objects, machinery, and waste materials [9,12]. Adjusted analyses have shown that workers aged 18–24 years, those who do not use personal protective equipment, those using stimulants like khat chewers, alcohol consumers, and those experiencing job dissatisfaction all face significantly higher odds of sustaining work-related injuries (adjusted odds ratios [aOR] = 1.72–2.32) [13]. These injuries directly impact production: injured workers often take sick leave, and some may even lose their jobs if disabled [14]. For instance, about 17% of waste collectors in Southern Ethiopia reported injuries that led to more than three days of work absence [13].

South Africa’s rising unemployment rate, now at over 30%, has led many into poverty [15], with some potentially finding work in buy-back centres handling recyclable waste. While these centres offer employment and income, the growing number of formal centres calls for investigating and managing waste recyclers’ work conditions and occupational health risks. Although existing literature has emphasised the sector’s contribution to waste management, pollution reduction, and job creation [5,7], there is a lack of quantitative data on injury patterns and risk factors among BBC workers. This gap underpins our study’s objective: to provide evidence-based estimates of injury prevalence and associated determinants to guide policy and intervention development.

Several studies have described the health outcomes of landfills and street recyclers [10,16]; however, we found no epidemiological studies on non-fatal workplace injuries among BBC workers in South Africa, particularly in Johannesburg. This study examines the prevalence and frequency of these injuries among waste recyclers at BBCs in Johannesburg, South Africa.

## 2. Materials and Methods

### 2.1. Study Design and Setting

This cross-sectional analytical study was conducted at buy-back recycling centres in Johannesburg, South Africa, between May and June 2021. The City of Johannesburg has a buy-back centre network that facilitates collecting and recycling materials such as glass, paper, and plastic. The number of buy-back centres fluctuates as new ones open and others close. For this study, ten buy-back centres in the City of Johannesburg were randomly selected from the municipality list.

### 2.2. Study Population and Sample

The study population comprised 286 formally employed waste recyclers aged 18 years and above at the ten buy-back centres. A final sample of 160 participants was estimated to achieve a 95% confidence level, with a ±7.75% margin of error. The sample size was calculated using a conservative estimate of the population proportion (*p* = 0.5) and a two-sided test at the 5% level of statistical significance. A list of employees was obtained from the buy-back centres, and study participants were selected using a simple random sampling strategy from the combined list of all employees.

### 2.3. Data Collection Tools and Methods

Trained fieldworkers provided potential participants with study information sheets, and those who gave written informed consent were enrolled. A structured, interviewer-administered questionnaire in English—with keywords translated into isiZulu and Sesotho, the most common local languages—was used to conduct interviews. The questionnaire items were adapted from a validated structured questionnaire previously used in a study on ‘health and health care access of landfill waste pickers in Johannesburg, South Africa’ [16]. Data from the interviews were captured directly onto REDCap version 13.10.4 using password-protected tablets. REDCap is a secure web application used to create, collect, and manage research data [17].

The prevalence of self-reported workplace injuries was derived from the following question: “In the last 6 months, have you experienced this type of injury at work?” (a) fractures, (b) sprains and muscle strains, (c) injured by the vehicle, (d) burns, (e) being hit by falling objects, (f) slips, trips, and falls, (g) cuts and lacerations, (h) inhaling toxic fumes, or (i) exposure to sudden loud noise. Based on the responses, the prevalence was dichotomised as ‘1’ if the participants said ‘yes’ to any injury listed above; otherwise, it was ‘0’. For the outcome variable, we dichotomised the frequency of injuries in the last 6 months into two categories: infrequent (combining participants not injured and those who reported being injured only a few times in the past 6 months) and frequent (combining participants who reported being injured several times a week and once or twice a month).

We considered nine exploratory variables, namely gender, age, highest level of education, tobacco smoking status, alcohol use, use of personal protective equipment (PPE), history of chronic diseases, hearing and or vision problems, and risk of mental health problems. We selected these variables based on existing literature, which highlights their relevance in occupational health research. Gender was categorised into female or male, with level of education categorised as no schooling, primary school, secondary school, or tertiary school. Tobacco smoking and alcohol use were classified as either yes or no. Participants who responded “yes” to using at least one PPE (mask, gloves, boots, or earplugs) were classified as using PPE. Similarly, those who reported at least one chronic disease (diabetes, hypertension, asthma, cancer, or stroke) were categorised as “yes” for a history of chronic disease. Lastly, those who reported being diagnosed with hearing or vision problems were categorised as “yes”. Based on the World Health Organisation Self-Reported Questionnaire tool for mental health [18], participants were classified as positive for mental health risk if they answered “yes” to 8 or more of the 20 questions. The cutoff of eight was chosen as validation studies showed it to yield optimal sensitivity and specificity [18].

### 2.4. Statistical Analysis

All statistical analyses were conducted using Stata version 19.5 (College Station, TX, USA). Descriptive statistics were used to summarise and describe study participants’ socio-demographics and occupational characteristics, as well as the types of self-reported workplace injuries experienced in the last 6 months. We used frequencies and percentages to describe categorical variables, and median, interquartile range (IQR), minimum, and maximum values to summarise continuous variables. Given that odds ratios are known to overestimate the effect size when the outcome is common (baseline prevalence > 10%) [19] and that log binomial models did not converge for all covariates, we fitted robust (modified) Poisson regression with robust variance estimation to report crude and adjusted relative risks (aRRs) and 95% confidence intervals (95% CIs) of factors associated with frequent self-reported workplace injuries. Forward stepwise model building was used to derive the final model, with predictors added to the model if they had a *p*-value  <  0.2. In the final model, the level of statistical significance was considered at 5%. We determined the model’s goodness-of-fit.

### 2.5. Ethical Considerations

Permission to conduct the study was obtained from the owners of each buy-back centre. No person-identifying information, such as names or addresses, was collected during the interviews. To ensure confidentiality, each participant was assigned a unique identifier. Ethical approval for the study was granted by the Human Research Ethics Committee (HREC) of the Faculty of Health Sciences, University of the Witwatersrand (Certificate No. M200266).

## 3. Results

More than half of the waste recyclers in this study were female (n = 89; 55.6%). The median age of the study participants was 32 years (IQR: 12; 18–74 years), and they had worked at the buy-back centres for a median of 1 year (IQR: 2; range 1–15 years). The median hours worked per day were eight (IQR 1; range 4–10 h), with a median monthly salary of ZAR 1450 (~USD 81.38) (IQR: ZAR 950.0; range 200–4000). Most respondents (n = 114; 71.2%) had completed secondary school, and two had tertiary education (n = 2; 1.2%). A quarter of the participants were current smokers (n = 39; 24.4%), and 32.5% reported alcohol use (n = 52). Less than a tenth had a history of chronic diseases (n = 13; 8.1%), and sensory impairments—hearing and/or vision problems—affected 15.0% (n = 24) of the study participants. A total of 11 participants (6.9%) were at risk for poor mental health. Almost all waste recyclers reported use of PPE (90.0%), yet 69.4% (n = 111) had at least one self-reported workplace injury in the prior six months. Regarding the frequency of injuries over the same period, 38.7% (n = 62) reported no injuries or only a few incidents, while 61.3% (n = 98) sustained injuries once or twice a month or several times a week (Table 1).

Table 2 details the self-reported injuries among the 111 participants injured at work during the previous six months. Among those who reported an injury at work, 90% (100/111) reported use of PPE. Cuts and lacerations were the predominant injury, reported among 67.6% of the participants (n = 75), followed by sprains or muscle strains (n = 44; 39.6%). Exposure to sudden loud noise was reported by 27.0% (n = 30), while 18.0% reported being hit by falling objects (n = 20). Approximately 15% reported experiencing slips, trips, or falls (n = 17; 15.3%), and being burnt at work (n = 16; 14.4%). Inhaling toxic fumes (n = 13) and vehicle-related injuries (n = 10) were less common (11.7% and 9.0%, respectively). No fractures were reported among study participants (Table 2).

In crude analyses, each additional year of age (RR 1.01; 95% CI: 1.00–1.02), history of chronic diseases (RR 1.43; 95% CI: 1.09–1.87), hearing or vision problems (RR 1.45; 95% CI: 1.15–1.83), or mental health risk (RR 1.37; 95% CI: 1.01–1.87) were significantly associated with frequent self-reported workplace injuries (Table 3). In the multivariable model, each additional year of age increased the relative risk of frequent self-reported workplace injuries by 1% (aRR 1.01; 95% CI: 1.00–1.02). Similarly, for each additional hour worked per day, the risk of frequent workplace injuries increased by 22% (aRR 1.22; 95% CI: 1.04–1.42). Compared to participants who did not have hearing or vision problems, waste recyclers with either impairment were 1.45 times more likely to report workplace injuries frequently (aRR 1.45; 95% CI: 1.14–1.83). Vision or hearing problems were associated with the largest increase in risk of frequent self-reported workplace injuries and were relatively common at 15%. We observed that certain factors (e.g., chronic disease, mental health risk) lost statistical significance once included in the adjusted model. This attenuation likely reflects shared variance with other covariates—such as age, hours worked per day, or sensory impairments—which accounted for a portion of their univariable effect on injury risk.

## 4. Discussion

This study describes the prevalence and frequency of workplace injuries among waste recyclers at buy-back centres (BBCs) in Johannesburg, South Africa. The purpose of this research was to identify the specific occupational risks faced by BBC workers, inform targeted prevention strategies, and contribute novel evidence on non-fatal injury patterns in the buy-back recycling environment. We found a high self-reported prevalence of workplace injuries of 69.4% in the previous six months. Notably, this rate is similar to the 73.2% prevalence reported among solid waste management workers in India [11]. Furthermore, a South African cross-sectional study of landfill waste pickers observed that 82.7% had sustained injuries or cuts during work [16]. This 10% difference between the prevalence of injuries at BBCs and landfill sites likely reflects the more controlled, semi-formal intermediate environment, which offers designated sorting areas and better access to basic protective equipment. In contrast, landfill waste pickers operate in unregulated settings with greater exposure to sharp hazards.

The most common injuries reported in our study were cuts and lacerations (67.6%) and sprains or muscle strains (39.6%). Similarly, among informal waste recyclers in landfill sites, lacerations and muscle pain accounted for 83% and 29% of reported injuries, respectively [20]. Among household waste collectors in the Republic of Korea, lacerations only accounted for 8.9% of injuries [21]. These disparities suggest that site infrastructure, task routines, and the nature of materials handled critically shape injury profiles. For instance, BBCs may concentrate on manual sorting of glass and metal, exacerbating cut risks, whereas domestic collection involves lighter refuse but more repetitive lifting. To reduce cuts and lacerations, BBCs should equip workers with cut-resistant gloves and enforce weekly debris-clearing sweeps. Sprains and strains can be prevented by installing height-adjustable benches, two-hour task rotations, and daily on-site stretching sessions.

Among the 111 participants who reported an injury at work, 90% used some form of PPE, suggesting that the equipment is unsuited to their tasks or worn incorrectly. Hasanah et al. [22] observed that, despite good PPE knowledge, 80% of waste collectors used it poorly, often citing discomfort and restricted mobility. This underscores the need for participatory selection of PPE—engaging workers in choosing designs that balance protection and ergonomics. Specifically, establishing yearly PPE trial sessions where recyclers test different glove models, coupled with on-site “fit and function” workshops led by trained safety officers, can improve compliance and comfort.

In the unadjusted analysis, chronic disease and mental health risk were associated with frequently reporting workplace injuries. However, these associations did not persist after controlling for confounders, which differs from a systematic review of 33 study populations that found some increased risk of accidental occupational injury among participants with metabolic diseases (diabetes) or mental ill-health [23]. This attenuation may reflect that physical factors like manual handling demands and extended hours mediate the relationship between health status and injury incidence, highlighting the importance of comprehensive occupational health assessments. Practically, BBCs could partner with local clinics to offer quarterly health screening days—checking blood sugar, blood pressure, and mental well-being—and then tailor workload adjustments or rest-break schedules for workers with identified risks.

Although modest, our study found that with every one-year increase in age, the risk of frequent workplace non-fatal injury increased by 1%. Much research has examined the relationship between age and workplace injuries, with mixed results, but most studies focus on severity rather than reporting frequency. For example, a meta-analysis of 31 studies showed older workers had lower odds of non-fatal accidents (OR 0.854; 95% CI 0.758–0.963) but higher fatality odds compared to younger workers (OR 2.060; 95% CI 1.762–2.410) [24]. This result mirrors a systematic review of 62 studies, which found that fatal injuries were more common among older workers, and non-fatal injuries prevailed in younger workers, although 19% of studies reported the opposite [25]. Our analysis is the first to link increasing age with more frequent self-reported non-fatal injuries at BBCs, underscoring the need for age-tailored interventions—such as strength-building exercises, workload rotation, and enhanced monitoring of ergonomic practices—for older recyclers.

We observed a 1.22-fold increase in the risk of frequent injury for each extra hour worked, which aligns with evidence that longer work hours induce fatigue and elevate injury risk [26,27]. Low BBC income may drive recyclers to extend hours for higher earnings, exacerbating injury frequency. Addressing this requires policy measures that cap daily work hours—such as an 8 h shift limit with mandatory 15 min breaks every two hours tracked via simple time-attendance registers [28]—incentivise safety over volume through injury-free week bonuses, and explore supplemental income opportunities to reduce overwork.

Waste recyclers who were diagnosed with either a hearing or vision problem had a higher risk of frequent workplace injuries compared to those without either impairment. This agrees with a previous study in the United Kingdom that found a relationship between hearing and vision problems and the risk of workplace injury [29]. Similarly, a systematic review reported a positive association between hearing and vision impairment and occupational injury [23]. Practical measures may include targeted training to compensate for sensory deficits, and adaptive aids such as high-visibility markings, amplified auditory warnings, high-contrast tape along conveyor edges, audible countdown timers for shift changes, and annual on-site vision and hearing check-ups.

To our knowledge, this is the first systematic assessment of both prevalence and frequency of non-fatal injuries among waste recyclers at formal BBCs, enhancing the applicability of our findings to these controlled recycling environments. However, relying on self-reported data may have introduced underreporting bias: participants might downplay or omit minor injuries to avoid potential stigma or disciplinary consequences. In addition, many workers may underreport injuries for fear of unpaid leave or job loss during recovery, given their precarious income.

## 5. Conclusions

This study found a high prevalence and frequency of workplace injuries among waste recyclers at buy-back centres. Older age, longer working hours, and vision or hearing problems were associated with frequent workplace injuries. To reduce injury risk, workplaces should implement targeted accommodations for sensory impairments (high-contrast markings, amplified alerts) and ageing-related vulnerabilities through simple ergonomic modifications (adjustable tables), alongside stricter regulation of working hours. A comprehensive PPE assessment is also essential to ensure proper usage and adequate protection, reinforcing safety measures among waste recyclers. However, these results may underestimate true injury rates due to reliance on self-reported data and potential underreporting driven by stigma or fear of income loss, which could bias associations toward the null. Future research should conduct direct workplace observations, link self-reports with medical records to validate injury data, trial tailored PPE interventions in controlled studies, and establish evidence-based thresholds for daily work hours and PPE usage that optimise safety in BBC settings.

## Figures and Tables

**Table 1 ijerph-22-01348-t001:** Socio-demographic and occupational characteristics, and health status of study participants.

	Total (N = 160)
**Gender**	
Male	71 (44.4%)
Female	89 (55.6%)
**Age in years**	
Median (interquartile range)	32 (12.0)
Minimum, Maximum	18, 74
**Years of work**	
Median (interquartile range)	1 (2.0)
Minimum, Maximum	1, 15
**Hours worked per day**	
Median (interquartile range)	8 (1.0)
Minimum, Maximum	4, 10
**Monthly salary (ZAR)**	
Median (interquartile range)	1450 (950.0)
Minimum, Maximum	200, 4000
**Level of education**	
No schooling	8 (5.0%)
Primary school	36 (22.5%)
Secondary school	114 (71.2%)
Tertiary level	2 (1.2%)
**Currently smoke**	
No	121 (75.6%)
Yes	39 (24.4%)
**Alcohol use**	
No	108 (67.5%)
Yes	52 (32.5%)
**Chronic disease/s**	
No	147 (91.9%)
Yes	13 (8.1%)
**Hearing and or vision problems**	
No	136 (85.0%)
Yes	24 (15.0%)
**Mental health**	
Not at risk	149 (93.1%)
At risk	11 (6.9%)
**Use of PPE**	
No	16 (10.0%)
Yes	144 (90.0%)
**Injured at work in the last 6 months**	
No	49 (30.6%)
Yes	111 (69.4%)
**Frequency of injury in the last 6 months**	
No injuries, or a few times	62 (38.7%)
Once or twice a month, or several times a week	98 (61.3%)

**Table 2 ijerph-22-01348-t002:** Description of use of PPE and self-reported workplace injuries in the last 6 months.

	Total (N = 111)
**Use of PPE**	
No	11 (10.0%)
Yes	100 (90.0%)
**Fractures**	
No	111 (100.0%)
**Sprains and muscle strains**	
No	67 (60.4%)
Yes	44 (39.6%)
**Injured by a vehicle**	
No	101 (91.0%)
Yes	10 (9.0%)
**Burns**	
No	95 (85.6%)
Yes	16 (14.4%)
**Hit by falling objects**	
No	91 (82.0%)
Yes	20 (18.0%)
**Slips, trips, and falls**	
No	94 (84.7%)
Yes	17 (15.3%)
**Cuts and lacerations**	
No	36 (32.4%)
Yes	75 (67.6%)
**Inhaling toxic fumes**	
No	98 (88.3%)
Yes	13 (11.7%)
**Exposure to sudden loud noise**	
No	81 (73.0%)
Yes	30 (27.0%)

**Table 3 ijerph-22-01348-t003:** Factors associated with frequent self-reported workplace injuries.

Variable	Univariable Analyses RR (95% CI)	Multivariable Analyses aRR (95% CI)
**Gender**		
Male	1	
Female	1.02 (0.79; 1.31)	
**Age in years**	1.01 (1.00; 1.02) *	1.01 (1.00; 1.02) *
**Hours worked per day**	1.16 (0.99; 1.35)	1.22 (1.04; 1.42) *
**Currently smoke**		
No	1	
Yes	1.06 (0.81; 1.40)	
**Alcohol use**		
No	1	
Yes	0.92 (0.70; 1.21)	
**Chronic disease/s**		
No	1	1
Yes	1.43 (1.09; 1.87) *	1.24 (0.92; 1.68)
**Hearing and or vision problems**		
No	1	1
Yes	1.45 (1.15; 1.83) *	1.45 (1.14; 1.83) *
**Mental health**		
Not at risk	1	1
At risk	1.37 (1.01; 1.87) *	1.31 (0.96; 1.78)
**Use of PPE**		
No	1	
Yes	0.88 (0.61; 1.26)	

* *p* < 0.05; CI: confidence interval; RR: relative risk; aRR: adjusted relative risk; PPE: personal protective equipment.

## Data Availability

The data supporting this study’s findings are available upon request from N.N.

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
