# Peer review of "Prevalence and Frequency of Non-Fatal Workplace Injuries Among Waste Recyclers at Buy-Back Centres in Johannesburg, South Africa: A Cross-Sectional Study"

_ijerph, 2025, doi:10.3390/ijerph22091348_

Round 1

Reviewer 1 Report

Comments and Suggestions for Authors

It is my pleasure to review the paper. After reviewing the manuscrpit, I have the following comments:

Introudction:

  • Correctly identify an important occupational health problem which worths further investigation.
  • Concisely and clearly explained the process of waste recycling in South Africa.
  • Explained the social factors affecting/ increasing the vulnerability of the workers

Materials and Methods:

  • Methods are well described
  • Subjects recruitment well described
  • however, the outcome variable ( infrequent and frequent injuries) has not been defined well. Instead of "several", "few", I think an exact number should be used.
  • Also, I would like to see why these 9 exploratory variables are selected.
  • And have the authors considered assessing any "previous training for the job or OSH" as a risk factor to be studies

Results

  • The results are clearly presented

Discussion:

  • The authors have appropriately compared their results with previous studies.
  • They have also correctly pointed out the ineffectiveness of "PPE use" in their setting. However, if they can also give more details on the type of PPE used by the workers, the readers may have more insight on why the PPE failed to protect the workers.
  • The authors have found a relationship between injuries and longer work hours. Is it possible to propose an appropriate length of work hour from their data? 

Conclusion:

  • The conclusion is satisfactory. 
  • I think the authors should also make some suggestions for further research on the subject.

Reviewer 2 Report

Comments and Suggestions for Authors

The manuscript addresses an important topic, but several aspects require clarification and improvement.

Section 1. Introduction

Some parts repeat the same points about recycling processes and BBCs without adding new detail, these could be merged.

General phrases such as “comprehensive research is essential” need clarification. What kind of data is missing, and why does it matter?

The choice of Johannesburg is not explained. The introduction should briefly explain why this context is relevant.

The section would benefit from a clearer formulation of the problem, a defined objective, and a short explanation of the study’s intended contribution.

Section 2. Materials and Methods

Clarify whether the random sampling was stratified by centre or done from a combined list of all employees.

Specify whether the questionnaire items were adapted from validated tools, especially for injury categories and exposure variables.

Mention whether model diagnostics were performed (e.g., checking for collinearity or influential observations).

Section 3. Results

Table 2 is not introduced in the text. A short lead-in would help guide the reader.

Clarify the transition from univariable analysis to the multivariable model. Some variables were significant in univariable analysis but lost significance after adjustment. A brief explanation of this shift would improve the statistical modelling.

Section 4. Discussion

The Discussion section repeats numerical findings from the Results section without additional interpretation or analysis. These data points should be linked to potential mechanisms, contextual explanations, or preventive implications to add value beyond descriptive reporting. The section should be revised to remove descriptive repetition and focus exclusively on the interpretation of the findings.

The manuscript does not state the study’s purpose, practical relevance, or scientific contribution. Authors should clarify the rationale behind conducting this research, specify its intended practical application, and explain how it contributes to existing knowledge, including identifying the risks workers are exposed to, the effects of these risks on workers, and the proposed measures to address them.

Indicate explicitly what is new compared to previous research.

The discussion lists limitations but does not suggest how future studies could address them. Specific research directions are recommended.

Section 5. Conclusions

The scientific contribution and the context of its practical relevance are missing. It is unclear why the study was important.

The formulation of recommendations is vague, for instance, terms such as “targeted accommodations” or “comprehensive PPE assessment” are not explained or exemplified.

Comments on the Quality of English Language

I prefer not to comment on the language quality and recommend that it be reviewed by someone with expertise.

Reviewer 3 Report

Comments and Suggestions for Authors

This is an interesting study, and the manuscript is generally well-structured and well-written. The article is recommended for publication after the following minor revision.

Suggestions for Authors:

Line 221-223: Although 90% of participants used some form of PPE, seven out of 10 workers reported an injury within six months, suggesting that the equipment is unsuited to their tasks or worn incorrectly.

As Table 1 indicates that 10% of participants were not using PPE. While in Lines 221-223, the phrase “7 out of 10” need to be specified as either a numerical value or a percentage, and it need to be clear whether it refers to PPE users or non-users, particularly in light of the authors’ suggestion that the equipment is either unsuitable for the tasks or improperly worn.

Round 2

Reviewer 1 Report

Comments and Suggestions for Authors

Thank you for the revisions made. I have no further comments on the manuscripts. 

Author Response

No further comments from the reviewer. 

Reviewer 2 Report

Comments and Suggestions for Authors

The manuscript has been substantially improved and addresses most of the concerns raised in the first review. Remaining issues are limited to the Discussion and Conclusions sections:

  • Section 4. Discussion: The practical relevance is still presented in general terms, without concrete or applicable examples. The authors should illustrate how the findings can be applied in practice, for example through interventions or policies tailored to the study setting.

  • Section 5. Conclusions: The authors should indicate how the study’s limitations affect the interpretation of the findings and propose concrete directions for future research.

Author Response

Response to reviewer comments (second round):

 Thank you for reviewing our manuscript. We have updated our manuscript and highlighted new changes in yellow.

  COMMENT 1.  Section 4. Discussion: The practical relevance is still presented in general terms, without concrete or applicable examples. The authors should illustrate how the findings can be applied in practice, for example, through interventions or policies tailored to the study setting.

Response:

We thank the reviewer for this suggestion. In the revised Discussion, we have embedded concrete, site-specific interventions and policies throughout Section 4 to translate each key finding into actionable steps for BBC operators and policymakers.

COMMENT 2. Section 5. Conclusions: The authors should indicate how the study’s limitations affect the interpretation of the findings and propose concrete directions for future research.

Response:

We appreciate this recommendation. In the revised Conclusion, we have explicitly acknowledged that reliance on self-reported injury data may lead to underestimation and attenuated associations. We have also added specific future research directions, including direct workplace observations, medical record linkage, controlled PPE intervention trials, and determination of optimal work-hour and PPE thresholds for buy-back centre environments. These changes address how our study’s limitations influence interpretation and outline concrete next steps.